# Effectiveness and Predictors of Long-Term Treatment Response to Tofacitinib in Rheumatoid Arthritis Cohort: General Analysis and Focus on High-Cardiovascular-Risk Subgroup—A Multicenter Study

**DOI:** 10.3390/medicina60121982

**Published:** 2024-12-02

**Authors:** Marta Priora, Andrea Becciolini, Eleonora Celletti, Myriam Di Penta, Alberto Lo Gullo, Marino Paroli, Elena Bravi, Romina Andracco, Valeria Nucera, Francesca Ometto, Federica Lumetti, Antonella Farina, Patrizia Del Medico, Matteo Colina, Viviana Ravagnani, Palma Scolieri, Maddalena Larosa, Elisa Visalli, Olga Addimanda, Rosetta Vitetta, Alessandro Volpe, Alessandra Bezzi, Francesco Girelli, Aldo Biagio Molica Colella, Rosalba Caccavale, Eleonora Di Donato, Giuditta Adorni, Daniele Santilli, Gianluca Lucchini, Eugenio Arrigoni, Emanuela Sabatini, Ilaria Platè, Natalia Mansueto, Aurora Ianniello, Enrico Fusaro, Maria Chiara Ditto, Vincenzo Bruzzese, Dario Camellino, Gerolamo Bianchi, Francesca Serale, Rosario Foti, Giorgio Amato, Francesco De Lucia, Ylenia Dal Bosco, Roberta Foti, Massimo Reta, Alessia Fiorenza, Guido Rovera, Antonio Marchetta, Maria Cristina Focherini, Fabio Mascella, Simone Bernardi, Gilda Sandri, Dilia Giuggioli, Carlo Salvarani, Veronica Franchina, Francesco Molica Colella, Giulio Ferrero, Alarico Ariani, Simone Parisi

**Affiliations:** 1Rheumatology Day Hospital and Outpatient Clinic, ASL CN1, 12100 Cuneo, Italy; marta.priora@gmail.com (M.P.); francesca.serale@gmail.com (F.S.); 2Internal Medicine and Rheumatology Unit, University Hospital of Parma, 43126 Parma, Italy; beccio@yahoo.it (A.B.); eleonoradidonato@ymail.com (E.D.D.); gadorni@ao.pr.it (G.A.); dsantilli@ao.pr.it (D.S.); glucchini@ao.pr.it (G.L.); dott.alaricoariani@libero.it (A.A.); 3Rheumatology Unit, Clinica Medica Institute, SS Annunziata Hospital, 66100 Chieti, Italy; myriamdipenta@gmail.com (M.D.P.); sabatiniemanuela3@gmail.com (E.S.); 4Rheumatology Unit, ARNAS Garibaldi, 95124 Catania, Italy; albertologullo@virgilio.it; 5Department of Clinical, Anesthesiological and Cardiovascular Sciences, Sapienza University of Rome, 00185 Rome, Italy; marino.paroli@uniroma1.it (M.P.); rosalba_caccavale@yahoo.it (R.C.); 6Reumathology Unit, Guglielmo da Saliceto Hospital, 29121 Piacenza, Italy; e.bravi@ausl.pc.it (E.B.); e.arrigoni@ausl.pc.it (E.A.); plate@ausl.pc.it (I.P.); 7Rheumatology Unit, Imperia Hospital, 18100 Imperia, Italy; r.andracco@gmail.com (R.A.); natalia.mansueto@libero.it (N.M.); 8Rheumatology Outpatient Unit, ASL Novara, 28100 Novara, Italy; v.nucera@asl.novara.it (V.N.); a.ianniello@asl.novara.it (A.I.); 9Rheumatology Outpatient Clinic, Azienda ULSS 6 Euganea, 35131 Padova, Italy; f.ometto@gmail.com; 10Rheumatology Unit, Azienda USL of Modena and AOU Policlinico of Modena, 41100 Modena, Italy; fedelumetti@gmail.com; 11Internal Medicine Unit, Rheumatology Outpatient Clinic, Augusto Murri Hospital, 63900 Fermo, Italy; antonella_farina@hotmail.com; 12Rheumatology Outpatient Clinic, Internal Medicine Unit, Civitanova Marche Hospital, 62012 Civitanova Marche, Italy; patdelmedico@inwind.it; 13Rheumatology Service, Internal Medicine Section, Department of Medicine and Oncology, Santa Maria della Scaletta Hospital, 40026 Imola, Italy; matteo.colina2@unibo.it; 14Alma Mater Studiorum, Department of Biomedical and Neuromotor Sciences, University of Bologna, 40126 Bologna, Italy; 15Rheumatology Unit, Santa Chiara Hospital APSS—Trento, 38122 Trento, Italy; viviana.ravagnani@gmail.com; 16Rheumatology Unit, Department of Medical Specialties, Nuovo Regina Margherita Hospital, 00154 Roma, Italy; palma.scolieri@gmail.com (P.S.); vinbruzzese@tiscali.it (V.B.); 17Division of Rheumatology, Department of Medical Specialties, La Colletta Hospital, ASL 3 Genova, 16132 Genova, Italy; maddalena.larosa@asl3.liguria.it (M.L.); dario.camellino@asl3.liguria.it (D.C.); gerolamo.bianchi@asl3.liguria.it (G.B.); 18Rheumatology Unit, Policlinico San Marco Hospital, 95121 Catania, Italy; elivisa21@gmail.com (E.V.); rosfoti5@gmail.com (R.F.); giorgioamato@hotmail.it (G.A.); francescodelucia89@yahoo.it (F.D.L.); yleniadalbosco@gmail.com (Y.D.B.); robertafoti@hotmail.com (R.F.); 19Rheumatology Unit, AUSL of Bologna—Policlinico Sant’Orsola—AOU—IRCCS of Bologna, 40138 Bologna, Italy; olga.addimanda@ausl.bologna.it (O.A.); massimo.reta@ausl.bologna.it (M.R.); 20Rheumatology Unit, ASL VC Sant’Andrea Hospital, 13100 Vercelli, Italy; rosetta.vitetta@aslvc.piemonte.it (R.V.); alessia.fiorenza@aslvc.piemonte.it (A.F.); guido.rovera.gr@gmail.com (G.R.); 21Rheumatology Unit, IRCCS Sacro Cuore Don Calabria Hospital, 37024 Negrar di Valpolicella, Italy; avolpe127@gmail.com (A.V.); antonio.marchetta@sacrocuore.it (A.M.); 22Internal Medicine and Rheumatology Unit, ASL Romagna—Rimini, 47924 Rimini, Italymariacristina.focherini@auslromagna.it (M.C.F.); fabio.mascella@auslromagna.it (F.M.); 23Rheumatology Unit, G.B. Morgagni—L. Pierantoni Hospital, 47121 Forli, Italy; francesco.girelli@auslromagna.it (F.G.); siiberna@yahoo.it (S.B.); 24Rheumatology Unit, Division of Internal Medicine, A.O. Papardo, 98158 Messina, Italy; aldomolica@alice.it; 25Rheumatology Department, Azienda Ospedaliero-Universitaria Città della Salute e della Scienza di Torino, 10126 Torino, Italy; fusaro.reumatorino@gmail.com (E.F.); mariachiaraditto@gmail.com (M.C.D.); simone.parisi@hotmail.it (S.P.); 26Rheumatology Unit, University of Modena and Reggio Emilia, 41125 Modena, Italy; gilda.sandri@unimore.it (G.S.); dilia.giuggioli@unimore.it (D.G.); carlo.salvarani@unimore.it (C.S.); 27Medical Oncology Unit, A.O. Papardo, 98158 Messina, Italy; verifra82@yahoo.it; 28Internal Medicine Unit, Milano-Bicocca University, 20126 Milano, Italy; francesco.molica3@gmail.com; 29Unit of Diagnostic and Interventional Radiology, Santa Corona Hospital, 17027 Pietra Ligure, Italy; giulio.ferrero@gmail.com

**Keywords:** rheumatoid arthritis, tofacitinib, cardiovascular risk

## Abstract

*Background and Objectives:* The treatment landscape for Rheumatoid Arthritis (RA) has evolved significantly with the introduction of Janus kinase inhibitors (JAKi), such as Tofacitinib (TOFA), which offer a new therapeutic option for patients who have failed or are intolerant to conventional synthetic disease-modifying antirheumatic drugs (csDMARDs). Safety concerns, particularly related to cardiovascular and cancer risks, prompted a need for additional investigation in real-world clinical settings. This study aimed to evaluate the long-term effectiveness and predictors of response to TOFA in two subpopulations of RA patients, categorized by differing cardiovascular risk profiles. *Materials and Methods:* This was a retrospective, multicenter observational study conducted as part of the BIRRA project, involving 23 Italian rheumatological referral centers. A total of 213 patients diagnosed with RA and treated with TOFA were included, with data collected on baseline demographics, clinical history, disease activity, and comorbidities. Patients were divided into high-risk and low-risk cardiovascular groups based on age (≥65 years) and the presence of at least one cardiovascular risk factor. Disease activity was assessed at baseline, 6 months, and 12 months using DAS28-ESR and DAS28-CRP. Treatment response was evaluated using intention-to-treat (ITT) and per-protocol (PP) approaches. Predictors of low disease activity (LDA) and remission were assessed through logistic regression, and clustering analyses were used to identify subgroups of patients with different therapeutic responses. *Results:* The study included 213 patients, with 129 classified as high-risk. For the overall cohort, patients achieving LDA and remission at 6 months were 20% and 12%, respectively, for the ITT analysis, and 29% and 14% for the PP analysis. At 12 months, 26% of patients reached LDA, and 17% achieved remission according to ITT, while for the PP analysis, these rates were 30% and 19%, respectively. No significant differences in remission or LDA rates were observed between the high-risk and low-risk groups. In the high-risk subgroup, 17% of patients reached LDA and 9% achieved remission at 6 months (ITT analysis), while these rates increased to 22% and 13%, respectively, in the PP analysis. At 12 months, 22% achieved LDA and 13% achieved remission in the ITT analysis, while 28% and 17% did so in the PP analysis. The reduction in DAS28-ESR and DAS28-CRP scores was significant (*p* < 0.001) across all time points for both high-risk and low-risk patients. Logistic regression analyses revealed that none of the baseline characteristics—including age, sex, comorbidities, rheumatoid factor, anti-citrullinated protein antibody (ACPA) positivity, initial disease severity, or treatment history—were significant predictors of remission or LDA at 6 or 12 months. The clustering analysis suggested that older patients, particularly those with worse baseline DAS28 scores, tended to show a less favorable response to treatment, potentially indicating impacts of age-related factors such as immunosenescence on therapeutic outcomes. *Conclusions:* Tofacitinib demonstrated similar effectiveness in both high- and low-risk cardiovascular subgroups of RA patients, with significant reductions in disease activity observed at both 6 and 12 months. Despite safety concerns related to cardiovascular risk, TOFA remained an effective treatment option across patient subgroups, with no significant differences in remission or LDA rates based on cardiovascular risk profiles. Age appeared to negatively impact treatment response, highlighting the role of immunosenescence in RA management. These findings support the use of TOFA as a personalized therapeutic option for RA, emphasizing the need for careful evaluation of cardiovascular and age-related risks in clinical decision-making.

## 1. Introduction

Rheumatoid arthritis (RA) is a chronic disease that belongs to the group of adult chronic inflammatory rheumatisms that lead to a progressive disability. If left untreated, rheumatoid arthritis can cause joint erosions and deformities that can lead to severe disability. In addition, extra-articular manifestations of RA, such as interstitial lung disease, vasculitis, and lymphoma, can be very serious and even life-threatening. In 2019, 18 million people worldwide were living with rheumatoid arthritis. About 70% of people living with rheumatoid arthritis are women, and 55% are older than 55 years [1]. Treatment options available today for RA have been greatly increased compared to the first class of conventional synthetic disease-modifying antirheumatic drugs (csDMARDs). Therefore, aiming for remission with the ‘treat-to-target’ attitude has become realistically more achievable at the present time. The most recent therapeutic option landed in clinical practice to patients with RA are oral small-molecule Janus kinase inhibitors (JAKi). Tofacitinib (TOFA) was the first JAKi to be approved in 2012 by the US Food and Drug Administration (FDA) and in 2017 by the European Medicines Agency (EMA) for the treatment, with or without methotrexate (MTX), of moderate to severe active RA in adult patients who have inadequately responded to or cannot tolerate one or more csDMARDs [1].

TOFA, inhibiting several JAK subtypes, especially JAK3 and JAK1, acts on synovial JAK/STAT targets through JAK-mediated IFN and IL-6 signaling pathways, thus blocking the role of JAK in synovial responses from playing a therapeutic role in RA [2,3].

Both the European Alliance of Associations for Rheumatology (EULAR) and the American College of Rheumatology (ACR) recommend JAKi as a second- and third-line treatment for RA [4,5]. Randomized controlled trials (RCTs) involving JAKi for the treatment of RA patients have demonstrated favorable outcomes [6]. In a series of phase II and phase III RCTs, the efficacy of TOFA has been extensively evaluated, demonstrating the efficacy profile of these small-molecule oral agents, prescribed either as monotherapy or in combination with MTX or other csDMARDs [7,8].

After the FDA approval of TOFA, concerns over a potential increased risk of serious infections, cardiovascular events, and cancers were raised. This finding prompted the FDA to require a post-marketing head-to-head safety trial [9] comparing the risk of major adverse cardiovascular events and cancers in patients with RA between TOFA 5 and 10 mg twice a day, and a tumor necrosis factor inhibitor (TNFi) among adalimumab or etanercept. Patient inclusion criteria were age ≥ 50 years and a medical history with at least one cardiovascular risk factor [9,10] from the ORAL Surveillance study, which in 2021 caused the FDA to require boxed warnings for TOFA to include risk information on serious heart-related events, cancer, blood clots, and death [3]. This study identified a high-risk and low-risk tofacitinib population with different relative risk vs TNFi. In 2023, the study results caused restrictions by EMA on the use of most oral JAKis by specific patient populations [11]. Thus, a posthoc analysis of ORAL Surveillance [10] allowed two subgroups of patients to be identified: a high-risk population, including patients aged ≥ 65 years and with at least one cardiovascular risk; and a low-risk population, including patients younger than 65 without any risk of major adverse cardiovascular events.

Today, due to all these evolutions in the history of this drug, extensive observational research on JAKi in RA is expanding, with particular attention to real-world clinical settings [12,13,14,15]. The aim of the present study is to evaluate the effectiveness and the predictors of clinical response in a large multicenter observational retrospective cohort of patients with RA, treated with Tofacitinib, considering the two different subpopulations, with different relative cardiovascular risks. Thus, data concerning the efficacy of this drug, extrapolated considering the two easily identifiable and clinically practical populations, can further improve the individualized benefit/risk assessment and clinical decision making regarding the treatment with TOFA.

## 2. Materials and Methods

### 2.1. Patients

As a part of the BIRRA (BIologics Retention Rate Assessment) project, this retrospective study was designed to evaluate the effectiveness of TOFA in RA patients. The study was performed according to the Declaration of Helsinki principles, and it was approved by the local ethics committees. Patients with confirmed diagnosis of RA who received TOFA were enrolled from 23 Italian rheumatological referral centers. The enrollment period extended from April 2019 to April 2023 with a maximum observation period of one year. Clinical history, treatment history and RA disease activity at baseline were recorded. For each patient, the following baseline data were recorded: general characteristics (age, sex, body mass index—BMI), smoking habit, disease duration, positivity for rheumatoid factor—RF, anti-citrullinated protein antibody -ACPA positivity, swollen and tender joint count—SJC and TJC, ESR, CRP, patient Visual Analogue Scale, disease activity, line of treatment, concomitant csDMARD use, concomitant steroids use, steroids dose (PDN-Eq), prior bDMARD use, prior tsDMARD use, and concomitant relevant disease. Disease activity was assessed in all patients by calculating both the DAS28-ESR and DAS28-CRP. Relevant comorbidities considered included diabetes, dyslipidemia, history of major adverse cardiovascular events (MACE), cancer, and arterial hypertension.

### 2.2. Statistical Analysis

Descriptive statistics of clinical and laboratory demographic characteristics were provided as medians with interquartile range (IQR) or percentage. Response to TOFA was assessed with an “intention to treat” (ITT) and “per protocol” (PP) analysis. The use of ITT and PP analyses in this study on the treatment of RA patients with TOFA is valuable for providing a comprehensive and accurate assessment of treatment efficacy. The ITT analysis includes all patients originally enrolled in the study, regardless of whether they completed the treatment or strictly adhered to the study protocol. This approach reflects the effectiveness of the treatment in real-world clinical practice, where some patients may discontinue or modify their treatment. In contrast, the PP analysis considers only those patients who completed the treatment as per the planned protocol, offering an estimate of the maximum efficacy of the treatment under ideal conditions. Using both analyses allows for an evaluation of both the real-world impact of the drug on the general population and its potential efficacy in patients with optimal adherence to the treatment.

DAS28-ESR/DAS28-CRP values at baseline, after 6 and 12 months of treatment were compared by Wilcoxon’s test and for paired nonparametric variables.

Logistic regressions tested which of the following factors were associated with achieving remission and/or LDA at 6 months and 12 months: age, sex, body mass index, smoking habit, duration of disease, presence of RF and/or ACPA, line of treatment, concomitant treatment with both steroids and csDMARDs, and DAS28-ESR/DAS28-CRP at baseline.

We also analyzed a subgroup of patients (high-risk group) who had at least one relevant comorbidity/risk factor among those considered by EMA (Table 1), including diabetes, dyslipidemia, hypertension, history or family history of coronary heart disease, history of venous thromboembolism, or history of cancer. The study was designed to focus on the extrapolation of data related to the high-cardiovascular-risk population, defined by the presence of at least one cardiovascular risk factor or age ≥ 65 years. While data from the overall group were used as a broader reference context, the primary aim was to specifically evaluate the characteristics and therapeutic response of this subgroup, without direct comparison to the low-risk population.

We used logistic regression to identify predictor variables of low disease activity and/or remission and clustering analysis to identify potential subgroups of patients with characteristics that influence therapeutic response.

All tests performed were two-tailed, and the significance level was set at 0.05 (with a 95% confidence interval).

## 3. Results

A total of 213 patients were included in the study. Their baseline characteristics are shown in Table 2.

Patients in LDA and in remission at 6 months were 20% and 12%, respectively, for the ITT analysis and 29% and 14%, respectively, for the PP analysis (Figure 1). Patients with LDA and in remission at 12 months were 26% and 17%, respectively, for the ITT analysis and 30% and 19%, respectively, for the PP analysis (Figure 1). There was no significant difference between the two analysis groups.

The analysis of the population considered at risk (129 patients; Table 2) according to the factors highlighted by EMA (Table 1) did not highlight substantial differences compared to overall patients, except for age (*p*: 0.000) and DAS28CRP (*p*: 0.000). No significant differences emerged either in terms of remission rate or in LDA.

Patients considered at high risk in LDA and in remission at 6 months were 17% and 9%, respectively, for the ITT analysis and 22% and 13%, respectively, for the PP analysis (Figure 2). Patients considered at high risk in LDA and in remission at 12 months were 22% and 13%, respectively, for the ITT analysis and 28% and 17%, respectively, for the PP analysis (Figure 3). There was no significant difference between the two analysis groups.

In the overall group, a significant reduction (*p* < 0.001) was observed in DAS28-ESR and DAS28-CRP (DAS28-ESR ΔT0–T1: 2.3; ΔT0–T2: 2.6 and DAS28-CRP ΔT0–T1: 2.5; ΔT0–T2: 2.7) at both time points (T1 and T2) (Figure 3).

Similarly, in the high-risk group, a significant reduction (*p* < 0.001) was observed in DAS28-ESR and DAS28-CRP (DAS28-ESR ΔT0–T1: 1.9; ΔT0–T2: 2.2 and DAS28-CRP ΔT0–T1: 2.1; ΔT0–T2: 2.6) at both time points (T1 and T2) (Figure 3).

Logistic regressions showed no predictors of remission or LDA at 6 months or 12 months. In particular, sex, age, seropositivity, presence of comorbidities, initial ESR and CRP values, line of therapy, association with csDMARDs and/or steroids did not show any significant correlation.

The clustering analysis highlighted the presence of a small group of patients, markedly older, who tend to have a worse DAS28 trend than the remainder of younger patients. In this group of patients, age is the variable that has the greatest impact on the improvement of the DAS28 among the patients in the cluster (*p* = 0.023). The other variables do not show a significant impact. Age is therefore moderately negatively correlated with improvement in DAS28. The results also highlighted that older patients tend to have a longer disease duration, but the data do not correlate with the number of treatment failures (*p* > 0.05). Thus, the duration of the disease does not seem to reflect (or in any case itself alone) a prolonged lack of therapeutic response over time.

However, the presence of multicollinearity and the limited number of observations suggest the need for additional data to obtain more robust results.

## 4. Discussion

In this study, a significant reduction in disease activity, assessed with DAS28-ESR and DAS28-CRP, was reported both at 6 and 12 months in RA patients after the start of therapy with TOFA.

The analysis of the subpopulation considered at higher risk according to the factors highlighted by EMA did not highlight substantial differences compared to the low-risk population, neither in terms of difference in reaching LDA nor in remission rate according to other data in the literature referring to other JAK inhibitors such as baricitinib [16].

None among the predictor factors assessed in this study (sex, age, seropositivity, presence of comorbidities, initial ESR and CRP values, line of therapy, association with csDMARDs and/or steroids), has shown a correlation with remission or LDA rates at 6 months or 12 months, in any of the two relative-risk subpopulations.

If a careful evaluation of cardiovascular risk is essential with regards to safety of TOFA, this humble study seems to bear witness to how this evaluation may not be equally valid for the evaluation of the effectiveness of the drug, since it resulted in a valid therapeutic option in both risk subgroups of the population examined.

To date, following the increasing attention to cardiovascular risk profiling with JAKi therapy, there are growing studies concerning many different safety aspects of the drugs [17,18,19].

Not so frequent are studies which, considering the profiling of cardiovascular risk and the clinical division into high- or low-risk subgroups, verify the possible differential rates of effectiveness.

We considered worthy of reflection a result highlighted by the clustering analysis, whereby a small group of patients with a worse DAS28 trend correlates with older age. Contextually, all the other variables do not show a significant impact. The presence of multicollinearity and the limited number of observations do not justify considering this data as having a strong scientific meaning; in any case, these results lead us to some reflections and insights.

Over the last decade, evidence has accumulated that, with progressive age, the immune system and the propensity for abnormal immunity change fundamentally, thus increasing the risk of infection, cancer and immune-mediated tissue damage. Senescence-related changes in innate and adaptive immune responses observed in healthy adults > 50 years of age are found to be anticipated and accelerated in chronic inflammatory conditions [20,21]. RA has been considered a model of premature senescence because, as the disease progresses, there is a higher prevalence of age-related diseases.

Senescent immune cells acquire the “senescence-associated secretory phenotype” which promotes and sustains tissue inflammation and is defective in balancing cytoplasmic kinase and phosphatase activities, changing their activation thresholds, weakening DNA repair and compromising the telomeric maintenance [22,23]. More than just epiphonema, in RA populations, these changes have been implicated in poor disease outcomes: they may contribute to the aggravation of both articular and extra-articular manifestations and may implicate a reduced response to standard treatments [24,25,26].

This study has some limitations that should be considered when interpreting the results. The retrospective observational design may introduce selection bias and limit the ability to establish direct causal relationships between treatment and outcomes. Data collection from multiple centers could lead to some heterogeneity in patient management. Finally, the relatively limited number of patients may reduce the statistical power of certain analyses, necessitating further prospective studies to confirm and expand upon the observed results.

## 5. Conclusions

In conclusion, the rheumatological population, as well as the global one, has been experiencing an aging trend. Results like those of the present study, by which the effectiveness of a drug seems to negatively correlate with age, lead us to imagine a deep interconnection between systemic inflammation, immunosenescence and age-related disease, worthy of consideration in those RA patients difficult to treat.

## Figures and Tables

**Figure 1 medicina-60-01982-f001:**
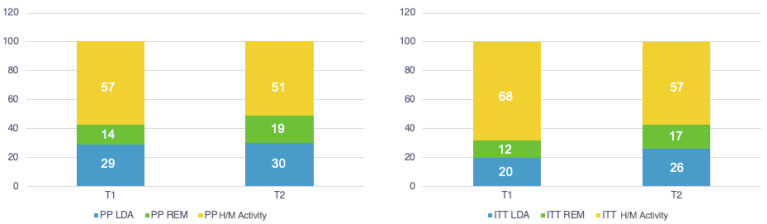
Patients achieving remission (REM), low disease activity (LDA), high/moderate (H/M) disease activity (Percentage) at T1 (6 months) and T2 (12 months). Intention to treat (ITT) and per protocol (PP) analysis.

**Figure 2 medicina-60-01982-f002:**
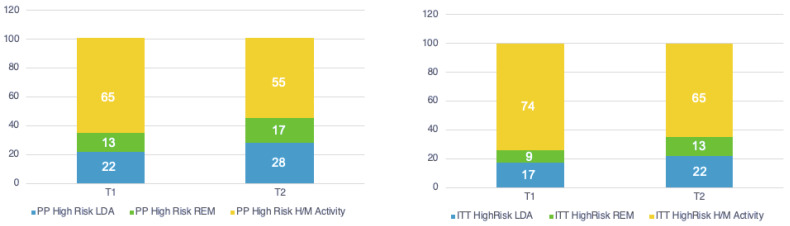
High-risk patients achieving remission (REM), low disease activity (LDA), high/moderate (H/M) disease activity (Percentage) at T1 (6 months) and T2 (12 months). Intention to treat (ITT) and per protocol (PP) analysis.

**Figure 3 medicina-60-01982-f003:**
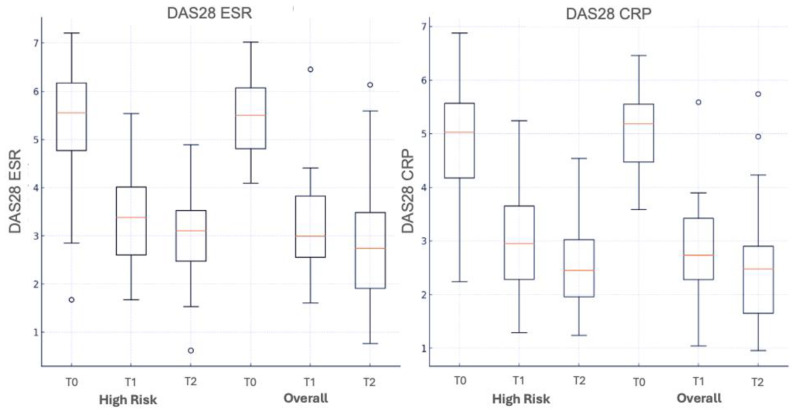
Comparison of the DAS28 ESR and DAS28 CRP trend between a group of high-risk patients (according to EMA) and overall at T0 (baseline), T1 (6 months) and T2 (12 months).

**Table 1 medicina-60-01982-t001:** Risks of major adverse cardiovascular events considered by EMA’s Pharmacovigilance Risk Assessment Committee (PRAC) [11].

Aged 65 years or older Long-term smokers or former smokers History of hypertension History of coronary heart disease Family history of coronary heart disease Dyslipidemia History of diabetes mellitus History of venous thromboembolism History of cancer

**Table 2 medicina-60-01982-t002:** Baseline characteristics split between the overall patient population and high-risk patients.

Characteristics		Overall Group	High-Risk Group
N		213	129
M:F n (%)		39 (18.3):174 (81.7)	12 (9.3):117 (90.7)
Age, median [IQR] yrs		60 [51–82] *	72 [61–86] *
Smokers, n (%)	Yes	42 (19.7)	23 (17.8)
	Former	44 (20.6)	22 (17.1)
	No	127 (66.3)	84 (65.1)
Body mass index, median [IQR] kg/m^2^		24.8 [22.0–42.7]	24.3 [21.0–39.7]
Disease duration, median [IQR], months		123 [78–532]	158 [82–402]
RF positivity, n (%)		140 (65.7)	81 (62.7)
ACPA positivity, n (%)		131 (61.5)	80 (62)
SJC, median [IQR]		6 [4–26]	5 [4–27]
TJC, median [IQR]		4 [2–22]	4 [2–24]
ESR, median [IQR], mm/h		32 [19.5–98]	36 [21–75]
CRP, median [IQR], mg/dL		1.4 [0.5–19]	1.9 [0.6–20]
VAS patient (0–100), median [IQR]		70 [50–100]	65 [45–100]
DAS28 (ESR), median [IQR]		5.3 [4.2–7.0]	5.6 [2.8–7.2]
DAS28 (CRP), median [IQR]		5.3 [3.6–6.6] *	5.0 [2.3–6.9] *
Line of treatment, [IQR]		2 [1–9]	2 [1–8]
Concomitant csDMARD use, n (%)	MTX	54 (25.3)	12 (19.4)
	LFN	4 (1.9)	3 (4.8)
	SSZ	2 (0.9)	0 (0.9)
	HCQ	10 (4.7)	3 (4.8)
Concomitant steroids use, n (%)		104 (48.8)	56 (43.4)
Steroids dose (PDN-Eq), median, mg/die		4 [5–25]	4 [5–25]
Prior bDMARD use, n (%)	TNFi	110 (50.2)	63 (48.9)
	IL6i	50 (23.5)	31 (24.0)
	IL1i	3 (1.4)	1 (0.8)
	CD20i	16 (7.5)	11 (8.5)
	CD80i	34 (16.0)	23 (17.8)
Prior tsDMARD use, n (%) JAKi-naive, n (%)		60 (28.2) 58 (27.2)	46 (35.6) 16 (12.4)
Concomitant relevant disease, n (%)	Diabetes	12 (5.6)	6 (9.7)
	Hypercholesterolemia	39 (18.3)	16 (25.8)
	MACE	9 (4.2)	6 (9.7)
	Arterial hypertension	61 (28.6)	19 (30.6)
	Cancer	11 (5.1)	3 (4.8)

RF, rheumatoid factor; ACPA, anti-citrullinated protein antibodies; SJC, swollen joints count; TJC, tender joints count; ESR, erythrocyte sedimentation rate; CRP, C-reactive protein, VAS, visual analogic scale; DAS28, disease activity score 28; csDMARD, conventional synthetic modified antirheumatic drugs; PDN-Eq, prednisone equivalent; bDMARD, biological modified antirheumatic drugs; tsDMARD, targeted synthetic modified antirheumatic drugs; TNFi, TNF inhibitors; IL-6i, IL-6 inhibitors; IL-1i, IL-1 inhibitors; CD20i, CD20 inhibitors; CD80i, CD80inhibitors; MACE, major adverse cardiovascular events. * significant difference.

## Data Availability

The data presented in this study are available on request from the corresponding author. The data are not publicly available due to privacy.

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
