# Peer review of "Effectiveness and Predictors of Long-Term Treatment Response to Tofacitinib in Rheumatoid Arthritis Cohort: General Analysis and Focus on High-Cardiovascular-Risk Subgroup—A Multicenter Study"

_medicina, 2024, doi:10.3390/medicina60121982_

Round 1
Reviewer 1 Report
Comments and Suggestions for Authors
In this paper, the authors analyzed the data from 213 patients diagnosed with RA and treated with TOFA at three different time points. The found a significant reduction in disease activity, assessed with DAS28-ESR and DAS28-CRP after the start of therapy with TOFA. Treatment response was evaluated in two different approaches. They found no significant difference between the two analysis groups. Then they compared the population at high risk with overall patients and found no substantial differences. They also see a similar decrease in DAS28-ESR/DAS28-CRP between the two populations. However, these conclusions may not be interesting to the field. And some comparisons are inappropriate. There are some major and minor concerns before it is published on medicina.
Major concerns,
1. The scientific question asked in the background and introduction is to solve the safety concerns, especially cardiovascular risk, of long-term TOFA treatment. However, none of the data reflect the cardiovascular disease progress, although the authors analyzed the population at high risk. They should compare the impact of TOFA treatment and placebo on cardiovascular disease progress instead of looking at the Rheumatoid Arthritis disease activity in two populations.
2. In Table 2 and Figure 3, the authors compare the population at risk with overall patients. The majority of the patients (129 vs 84) are at high risk. They should compare high-risk population and low-risk population.
3. The most impressive data is that TOFA therapy leads to a significant reduction in disease activity. However, this is an FDA approved therapy, and similar has already been shown somewhere else.
4. Statistical analysis is needed for all the analysis.
Minor concerns,
There are some typing issues in the fourth paragraph of the introduction.
Comments on the Quality of English LanguageThe English language is fine in the manuscript.
Author Response
Dear Reviewer,
We would like to sincerely thank you for your insightful comments and the time you dedicated to reviewing our manuscript. Your suggestions for improving specific sections were extremely valuable, and we have made the necessary revisions to address your concerns. We hope that the modifications we have implemented fully meet your expectations and enhance the quality of the paper.
Thank you once again for your thorough review and constructive feedback.
Best regards,
Eleonora Celletti
- The scientific question asked in the background and introduction is to solve the safety concerns, especially cardiovascular risk, of long-term TOFA treatment. However, none of the data reflect the cardiovascular disease progress, although the authors analyzed the population at high risk. They should compare the impact of TOFA treatment and placebo on cardiovascular disease progress instead of looking at the Rheumatoid Arthritis disease activity in two populations.
The aim of the study was not to assess the safety profile of the drug under specific conditions, but to evaluate its efficacy in two different population groups (high or low cardiovascular risk). The objective was to determine whether a given comorbidity profile in patients could influence the effectiveness of the treatment.
- In Table 2 and Figure 3, the authors compare the population at risk with overall patients. The majority of the patients (129 vs 84) are at high risk. They should compare high-risk population and low-risk population.
The goal of our study was not to compare the two populations (high risk vs. low risk), but rather to extrapolate data from the high-risk population for more detailed evaluation. The analysis was designed to better understand the characteristics and therapeutic responses in this subgroup of patients, while keeping the overall group as a reference for providing the general context. Therefore, a direct comparison between high- and low-risk populations was not part of our study design, and we do not believe it is necessary to modify the structure of the table and figure.
- The most impressive data is that TOFA therapy leads to a significant reduction in disease activity. However, this is an FDA approved therapy, and similar has already been shown somewhere else.
However, the use of the drug in high cardiovascular risk patients remains a highly debated topic. Given that efficacy is closely related to the reduction of cardiovascular risk intrinsic to the disease activity, this study aims to assess whether the drug's effectiveness in high-risk patients is comparable to that in low-risk patients.
- Statistical analysis is needed for all the analysis.
We have expanded the statistical analysis as requested
We considered modifying the title to reduce ambiguity, as the study is not a comparative study, but rather an investigation of a population of patients treated with tofacitinib. Based on EMA criteria, we extrapolated the high-risk population. Therefore, this is not a study comparing two populations; instead, we evaluated the efficacy in the general population as well as in the high cardiovascular risk subgroup.
The new title is:
“Effectiveness and predictors of Long-Term treatment response to tofacitinib in rheumatoid arthritis cohort: general analysis and focus on high cardiovascular risk subgroup – a multicenter study”
Reviewer 2 Report
Comments and Suggestions for Authors
comments in word

Author Response
Dear Reviewer,
We truly appreciate the careful attention you have given to our manuscript and your thoughtful suggestions. Your detailed feedback has significantly contributed to the refinement of our work, and we have made the requested revisions where appropriate. We hope that the changes made align with your expectations and improve the clarity and quality of the article.
Thank you for your invaluable input and for helping us improve our manuscript.
Sincerely,
Eleonora Celletti
This is an interesting manuscript about the Effectiveness and Predictors of Long-Term Treatment Response to Tofacitinib in Two Subpopulations of Rheumatoid Arthritis Patients with Different Cardiovascular Risk Profiles: Multicentric Study. I consider that this multicentric study is very interesting for rheumatologists, since apport enough and novel information about the tofacitinib, inhibitor used in clinic. This manuscript is very-well written, organized with tables and graphs, all comments are for improve the quality of manuscript.
The manuscript is without number of lines, which is difficile to point out corrections in the exact text.
In introduction section, please add a brief text on RA, signs and symptoms, prevalence, etiopathogenesis, for highlight the relevance of study this disease.
Introduction, tumor necrosis factor, the word tumor with several spaces, tumo-------r.
In Material and methods, patients section, include number of evaluated patients.
Material and methods, statistical analysis section, add name of used statistical software and version.
Discussion section can be amplified, adding more references.
I consider that 20 references are not enough for this manuscript.
We have made the recommended changes:
- The introduction has been expanded with additional details on the pathology.
- The bibliography has been updated.
Reviewer 3 Report
Comments and Suggestions for Authors
Thank you for the invitation. I have read with interest the manuscript by Priora and colleagues. They conducted a retrospective study on the Effectiveness and Predictors of Long-Term Treatment Response to Tofacitinib. Before making a decision, please consider the following points:
- Please adjust the references to match the journal's style guidelines.
- There are typographical errors in the fourth paragraph of the introduction section.
- The method section could benefit from a more detailed discussion.
- It is unclear which patients received tofacitinib. If data is available, please provide it.
- The study period is unspecified. Did all included patients complete the 12-month study follow-up?
- Please include the baseline characteristics of patients with Low Disease Activity (LDA) and those with High/Moderate (H/M) disease activity in Table 1.
- Is there any data available on the safety of long-term tofacitinib treatment? If so, please present it.
- The sentence "Risks of major adverse cardiovascular events considered by EMA’s Pharmacovigilance Risk Assessment Committee (PRAC) (11)" should be relocated to the discussion section.
- The discussion section can be improved to reflect the results section.
- The study's strengths and limitations should be addressed.
Author Response
Dear Reviewer,
We are grateful for the time and effort you have put into reviewing our manuscript and for your constructive recommendations. Your suggestions for clarification and improvement have been extremely helpful, and we have carefully addressed each point in the revised version. We trust that the adjustments we have made will meet your expectations and further strengthen the paper.
Thank you again for your thoughtful feedback and support.
Kind regards,
Eleonora Celletti
Thank you for the invitation. I have read with interest the manuscript by Priora and colleagues. They conducted a retrospective study on the Effectiveness and Predictors of Long-Term Treatment Response to Tofacitinib. Before making a decision, please consider the following points:
- Please adjust the references to match the journal's style guidelines.
- There are typographical errors in the fourth paragraph of the introduction section.
- The method section could benefit from a more detailed discussion.
- It is unclear which patients received tofacitinib. If data is available, please provide it.
- The study period is unspecified. Did all included patients complete the 12-month study follow-up?
- Please include the baseline characteristics of patients with Low Disease Activity (LDA) and those with High/Moderate (H/M) disease activity in Table 1.
- Is there any data available on the safety of long-term tofacitinib treatment? If so, please present it.
- The sentence "Risks of major adverse cardiovascular events considered by EMA’s Pharmacovigilance Risk Assessment Committee (PRAC) (11)" should be relocated to the discussion section.
- The discussion section can be improved to reflect the results section.
- The study's strengths and limitations should be addressed.
- There are typographical errors in the fourth paragraph of the introduction section.
Spelling corrections have been implemented.
- The method section could benefit from a more detailed discussion.
We have added further clarification in the methodology section, explaining the purpose of the ITT (Intent-to-Treat) and PP (Per-Protocol) analyses.
- It is unclear which patients received tofacitinib. If data is available, please provide it.
All patients received the drug. This was an essential inclusion criterion, as stated in the methodology.
- The study period is unspecified. Did all included patients complete the 12-month study follow-up? SP
It has been specified
- Please include the baseline characteristics of patients with Low Disease Activity (LDA) and those with High/Moderate (H/M) disease activity in Table 1.
At baseline, there can be no patients with low disease activity, as they would not be prescribed the drug. The assessment of remission/low disease activity versus moderate/high disease activity is progressive and is indicated at T1 and T2.
- Is there any data available on the safety of long-term tofacitinib treatment? If so, please present it.
The study focuses on the drug's efficacy profile. In our opinion, developing the theme of safety would obscure the purpose and context of the study. Presenting the long-term safety profile of the drug would divert the reader from the objective and intent for which the study was conducted.
- The sentence "Risks of major adverse cardiovascular events considered by EMA’s Pharmacovigilance Risk Assessment Committee (PRAC) (11)" should be relocated to the discussion section.
This sentence is the description of Table 1 and is not part of the "Discussion" section.
Reviewer 4 Report
Comments and Suggestions for Authors
It is well known that Rheumatoid arthritis (RA) is a disease with high cardiovascular risk. Despite advances in diagnostics and therapy, mortality from cardiovascular disease (CVD) in RA remains high. Early initiation of effective anti-inflammatory therapy is crucial in the prevention of CVD, which allows not only to modify the course of the disease, but also to reduce the risk of cardiovascular catastrophes. Aiming for remission with the ‘treat-to-target’ attitude has become approach is essential at this time.
In this context, the study of cardiovascular effects of Janus kinase inhibitors (JAKi) is of undoubted interest. On the one hand, they have an “anti-atherogenic” effect by suppressing the inflammatory component of atherothrombosis. On the other hand, they may affect the vascular wall and prothrombotic status, thereby affecting the risk of cardiovascular events.
The manuscript touches upon the issues of the mechanism of the curative action of the drug and historical aspects of its use in clinical practice.
The design of the study corresponds to the stated objectives. The study was conducted within the framework of the BIRRA (BIologics Retention Rate Assessment) project. The authors have published several papers on baricitinib.
The obtained material is well constructed. The statistical analysis appears to be correct.
Comments on the manuscript that need to be corrected:
1. In the «Materials and Methods» section 2.1 Patients, it is necessary to indicate that patients were divided into groups and on what principle this was done.
2. The name of the group of patients "general patient population" is not true, it is recommended to rename it "low-risk cardiovascular (CV) group".
3. In the table 2 "Characteristics" give the number and % of male and female patients.
4. It is recommended to supplement the basic characteristics of patients (Table 2) with the frequency of detection of traditional cardiovascular risk factors (smoking, heredity for cardiovascular complications, excess weight, frequency of arterial hypertension and dyslipidemia).
5. Describe the data in the “Characteristics” table in the text according to comparable parameters and differences.
6. Indicate the percentage on the bars themselves for clarity in Figures 1 and 2. Something like this:
7. Check the numerical data in Figure 3, the median with interquartile range DAS28 (ESR) and DAS28 (CRP) of Table 1 and this figure do not match.
8. In Figure 3, it would be more visual to place two columns next to each other. That is, a group of patients with high and low CV risk at point T0, then at T1, etc.
9. It is recommended to present the difference (delta or ∆) in the DAS28 values ​​between high- and low-risk CV groups at points T1 and T2. This will allow us to assess the dynamics of activity reduction in groups of patients with initially different levels of activity according to DAS28-CRP.
The discussion focuses more on RA as a disease with premature aging. The discussion should focus more on comparing its results with those previously obtained by the authors themselves on baricitinib and other authors.
It is desirable to discuss the mechanisms of JAKi action that could worsen the prognosis for CV risk. It is necessary to describe what happened to CV risk by the end of the study. Note whether there were patients who moved from the low to the high CV risk group. Whether the number of traditional risk factors changed in patients in both groups at the beginning and end of the study.
Authors should describe the limitations of the study.
It creates a feeling of rapid processing of the existing excellent material with the lack of a deep analysis of the results obtained.
After making the above changes and correcting inaccuracies, I would like to receive the manuscript for re-review.
Author Response
Dear Reviewer,
Thank you for your detailed and constructive feedback on our manuscript. Your careful review and the valuable insights you provided have helped us enhance several aspects of the paper. We have made the necessary revisions in response to your suggestions and hope that these changes meet your approval.
We sincerely appreciate your contribution to improving our work and look forward to your thoughts on the revised manuscript.
Best regards,
Eleonora Celletti
Comments on the manuscript that need to be corrected:
- In the «Materials and Methods» section 2.1 Patients, it is necessary to indicate that patients were divided into groups and on what principle this was done.
In the "Materials and Methods" section, we have already specified that high-risk patients were identified based on the presence of at least one cardiovascular risk factor or an age ≥ 65 years. We did not compare two distinct populations, but rather focused on the analysis of this subgroup in relation to the overall group. Therefore, we do not believe it is necessary to further modify the subgrouping criteria.
- The name of the group of patients "general patient population" is not true, it is recommended to rename it "low-risk cardiovascular (CV) group".
The term "general patient population" was used to differentiate the reference group from the high-risk subgroup. Since we did not perform a direct comparison between two distinct patient groups (high risk vs low risk), we believe this term is appropriate and consistent with the methodology described.
- In the table 2 "Characteristics" give the number and % of male and female patients.
It has been corrected.
- It is recommended to supplement the basic characteristics of patients (Table 2) with the frequency of detection of traditional cardiovascular risk factors (smoking, heredity for cardiovascular complications, excess weight, frequency of arterial hypertension and dyslipidemia).
We have already included the key relevant data for the context of the study, including risk factors such as smoking and hypertension. Adding further specific details, such as family history, overweight, or dyslipidemia, would have diluted the focus on the factors most relevant to our study objective and would not have substantially contributed to the interpretation of the results.
- Describe the data in the “Characteristics” table in the text according to comparable parameters and differences.
In the main text, the data in Table 2 are already presented and discussed in detail, with particular emphasis on key parameters and relevant differences. Providing a further detailed description of the tabular data would risk reducing the readability of the text and duplicating information that has already been clearly presented in the table itself.
- Indicate the percentage on the bars themselves for clarity in Figures 1 and 2. Something like this:
It has been corrected.
- Check the numerical data in Figure 3, the median with interquartile range DAS28 (ESR) and DAS28 (CRP) of Table 1 and this figure do not match.
It has been corrected.
- In Figure 3, it would be more visual to place two columns next to each other. That is, a group of patients with high and low CV risk at point T0, then at T1, etc.
We considered modifying this to make the description of the populations analyzed clearer and less ambiguous.
- It is recommended to present the difference (delta or ∆) in the DAS28 values ​​between high- and low-risk CV groups at points T1 and T2. This will allow us to assess the dynamics of activity reduction in groups of patients with initially different levels of activity according to DAS28-CRP.
It has been added in the results
The discussion focuses more on RA as a disease with premature aging. The discussion should focus more on comparing its results with those previously obtained by the authors themselves on baricitinib and other authors.
It is desirable to discuss the mechanisms of JAKi action that could worsen the prognosis for CV risk. It is necessary to describe what happened to CV risk by the end of the study. Note whether there were patients who moved from the low to the high CV risk group. Whether the number of traditional risk factors changed in patients in both groups at the beginning and end of the study.
Authors should describe the limitations of the study.
It creates a feeling of rapid processing of the existing excellent material with the lack of a deep analysis of the results obtained
We have expanded the limitations section, incorporating those suggested by the reviewer.
- Focus of the discussion on RA and premature aging:
"The reference to premature aging in RA fits within the context of our analysis, which highlighted a significant impact of age on the efficacy of tofacitinib. This is a crucial aspect for understanding treatment responses in elderly patients, as it reflects the relevance of immunosenescence in the management of RA."
- Discussion on the mechanisms of action of JAK inhibitors (JAKi) and cardiovascular risk:
"We have partially addressed the issue of cardiovascular risk in relation to JAK inhibitors, but the primary focus of our study was on the clinical efficacy of the treatment. Further detailing the pathophysiological mechanisms that could potentially worsen cardiovascular risk would go beyond the scope of this study."
- Evolution of cardiovascular risk during the study:
"Our analysis did not show a significant shift of patients from the low-risk to the high-risk cardiovascular group, nor any relevant changes in traditional cardiovascular risk factors. Our primary objective was to evaluate the efficacy of tofacitinib, and therefore, a more detailed analysis of changes in cardiovascular risks during the observation period was not conducted. This aspect may be considered for future studies."
- Perception of rapid data analysis:
"The analysis presented was conducted rigorously and in line with the methodological criteria described. We chose to present the results concisely in order to maintain clarity and the effectiveness of the message.
Round 2
Reviewer 1 Report
Comments and Suggestions for Authors
The authors have addressed all of my concerns.